# Nature of excitations and defects in structural glasses

Camille Scalliet [ID] [1]*, Ludovic Berthier [ID] [1] & Francesco Zamponi[2]

The nature of defects in amorphous materials, analogous to vacancies and dislocations in crystals, remains elusive. Here, we explore their nature in a three-dimensional microscopic model glass-former that describes granular, colloidal, atomic and molecular glasses by changing the temperature and density. We find that all glasses evolve in a very rough energy landscape, with a hierarchy of barrier sizes corresponding to both localized and delocalized excitations. Collective excitations dominate in the jamming regime relevant for granular and colloidal glasses. By moving gradually to larger densities describing atomic and molecular glasses, the system crosses over to a regime dominated by localized defects and relatively simpler landscapes. We quantify the energy and temperature scales associated to these defects and their evolution with density. Our results pave the way to a systematic study of low-temperature physics in a broad range of physical conditions and glassy materials.

[1] Laboratoire Charles Coulomb (L2C), Université de Montpellier, CNRS, 34095 Montpellier, France. [2] Laboratoire de Physique de l'Ecole Normale Supérieure, ENS, Université PSL, CNRS, Sorbonne Université, Université de Paris, 75005 Paris, France. *email: cs2057@cam.ac.uk

Amorphous solids may be prepared via two distinct routes. Atomic and molecular glasses are obtained by crossing a glass transition, by cooling a dense liquid at constant pressure, or by a compression at constant temperature[1]. Disordered assemblies of grains, droplets, and large colloids solidify by crossing a jamming transition upon compression from a fluid state[2]. In a seminal work, Liu and Nagel[3] proposed a unified phase diagram for glass and jamming transitions. Subsequent work investigated amorphous solidification by interpolating continuously between these two limits[4], to understand similarities and differences between them.

Zero-temperature amorphous solids are minima of the many-body interaction potential, and their low-temperature properties are determined by the structure of the potential energy landscape around these minima[5,6]. The low-frequency vibrational modes around a minimum define the excitations of the solid (analogous to phonons in a crystal), and the structure of energy barriers separating nearby energy minima define the defects (analogous to vacancies and dislocations in a crystal). The nature of excitations and defects can be different in systems undergoing jamming or glass transitions.

Understanding the nature of excitations and defects in amorphous solids is one of the major goals of condensed matter physics, because these features are relevant for mechanical, thermal, and transport properties[6,7], such as specific heat and thermal conductivity[8,9], sound attenuation[10], energy dissipation[11] (with important technological consequences, e.g. for gravitational wave detection[12]), solid elasticity, and plasticity[13–17].

In dense atomic glasses, typically modeled by simple Lennard–Jones (LJ) interaction potentials, low-frequency excitations are phonon-like (with peculiar properties)[18–21], and defects are localized, with a few particles jumping between two local minima and slightly perturbing their neighbors, giving rise to two-level systems that play an important role in the low-temperature thermal properties of glasses[7,22]. This phenomenology has been numerically confirmed in simple LJ-like glass models[6,23,24].

A different situation holds near jamming where particles interact essentially via short-range repulsive forces and mechanical stability is controlled by particle contacts. The minimal number of contacts required for stability is the isostatic number, which is exactly reached at the jamming transition. In the vicinity of the transition, the solid is marginally stable[2,25]: a small perturbation can remove a few contacts and destabilize the entire solid[26,27]. This geometrical feature gives rise to low-frequency collective excitations that are extremely different from phonons[21,28,29]. The energy landscape features a large number of minima[30], separated by low barriers. The corresponding defects are thus extended and involve collective particle motion, as numerically observed in simulation of hard sphere (HS) glasses[31,32].

The mean-field theory of the glass transition has attempted to describe this phenomenology, coming to two important conclusions[30]. First, glass and jamming transitions are distinct phase transitions, where solidity emerges at thermal equilibrium (glass transition) or at zero temperature (jamming transition). Second, for repulsive particles, the jamming transition is buried deep inside a glass phase, which is split into two distinct phases, containing either trivial excitations ('simple glass'), or collective ones ('marginal glass'). Mean field theory predicts that a sharp Gardner phase transition separates these two glass phases[30]. These mean field results, however, conflict with several finite dimensional studies. First, the nature of the Gardner transition in three dimensions is not fully understood[33]. Second, the quasi-localized excitations[20,21] and localized defects[23,34] numerically observed in glassy systems are not described by mean-field theory.

In summary, the two extreme cases of LJ-like and HS glasses, which have been thoroughly investigated by numerical simulations, together with the insight coming from mean field theory, highlight the conceptually distinct nature of excitations and defects in glassy and jammed states. Yet, many questions are currently open. Are extended, marginally stable excitations restricted to HS glasses[35]? How do excitations and defects evolve between the jamming and glass regimes? What is the region where marginal stability influences glass properties? Do localized and collective excitations coexist in some regime?

We address this broad set of questions by introducing two main technical tools. First, we simulate a three-dimensional system of particles interacting via the Weeks–Chandler–Andersen (WCA) potential[36]. Varying a physical control parameter, i.e. the density, allows to capture all relevant glassy regimes. At high density, the WCA potential becomes equivalent to a Lennard–Jones potential, widely used for atomic glasses. At lower density, the finite range of the WCA potential makes it suitable to study jamming. Second, we perform a systematic investigation of the distribution of minima and barriers in the potential energy landscape, by means of a state-of-the-art reaction path finding protocol.

Our analysis leads to two main findings. First, the HS phenomenology extends in a wide region of the phase diagram of the soft WCA system, centered around the jamming transition. Marginal stability, and its associated collective excitations and defects, is then relevant not only for hard colloids, but also for softer systems such as foams and emulsions. Second, marginal stability can coexist with localized defects over a wide range of physical conditions. Collective modes are typically associated to low-energy barriers, whereas localized modes correspond to the motion of a few particles in a rigid elastic matrix, controlled by higher energy barriers. The two phenomena are thus controlled by different energy and temperature scales that we quantify numerically. In the vicinity of jamming, all defects are collective. At intermediate density, collective defects exist only under a characteristic temperature 'dome', as predicted by mean-field theory[17,37], while localized defects dominate at higher temperatures. At high density, collective defects disappear. Our results thus establish the extent of the region in which marginal stability impacts glass physics, and open the way for a systematic study of low-temperature glass physics (including quantum effects) across the whole range of experimentally relevant conditions.

## Results

**Equilibrium phase diagram.** We first discuss the packing fraction, $\varphi$, and temperature, $T$, equilibrium phase diagram of the three-dimensional polydisperse, non-additive WCA model we simulate (see the Methods section for technical details). We focus in particular on the determination of the fluid region, and its boundary, the glass transition, below which physical dynamics fail to reach equilibrium.

For each packing fraction, we study the temperature evolution of the relaxation time $\tau_\alpha$ of density correlations in the equilibrium fluid, using molecular dynamics (MD) simulations. The relaxation time is measured through the self-intermediate scattering function $F_s(k, t)$, by the condition $F_s(k = 7.0, t = \tau_\alpha) = e^{-1}$, and it follows an Arrhenius law at high temperature, while deviations from the Arrhenius law appear below an onset temperature $T_0$, where $\tau_\alpha(T = T_0) \equiv \tau_0$. Standard algorithms such as MD fail to reach equilibrium below $T_d$, defined by $\tau_\alpha(T = T_d) = 10^4 \tau_0$. We refer to $T_d$ as the "computer glass transition". Glasses prepared at the computer glass transition lie much higher in the energy landscape than any experimental glass, which are created after up to 12 decades of glassy dynamical slowdown.

In order to bypass the computer glass transition and access deeper glassy minima, relevant to describe real materials, we use a hybrid swap Monte Carlo algorithm[38]. This unphysical

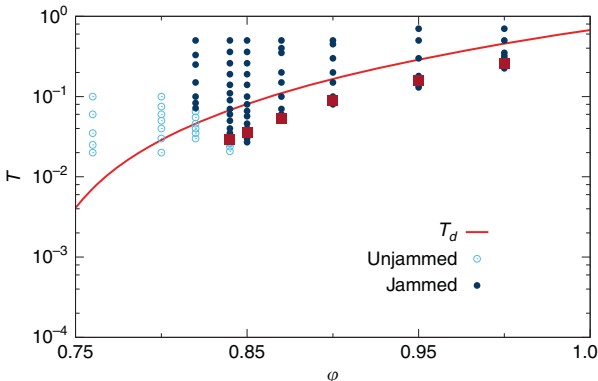

**Fig. 1** Equilibrium phase diagram of the WCA model. The red line corresponds to the computer glass transition, below which conventional molecular dynamics simulations fail to reach equilibrium over accessible time scales. State points are studied in equilibrium conditions with the hybrid swap method (circles). After energy minimization, their potential energy is either zero (open circles, unjammed), or positive (full circles, jammed). The equilibrium glassy states analyzed in more detail are indicated with red squares

dynamical scheme, which accelerates greatly equilibrium sampling[39,40], achieves thermalization down to temperatures around $0.6 T_d$ almost independently of the packing fraction.

In Fig. 1 we indicate by circles the state points for which equilibrium is reached, either by MD or swap. The state points below the computer glass transition line $T_d(\varphi)$ have been fully equilibrated by swap. In this article, we present results for glasses prepared well below the computer glass transition, highlighted by red squares in Fig. 1. They lie along a line $T_g \approx 0.65\, T_d$ of similar glass stabilities, which corresponds roughly to the experimental glass transition temperature. The latter is defined by $\tau_\alpha = 10^{12}\tau_0$, and estimated by extrapolating the physical relaxation time with a parabolic law. These glasses lie so deep in the landscape that when their physical dynamics is simulated with MD, diffusion is suppressed over accessible timescales, and only vibrations around the initial equilibrated configuration are observed. The landscape picture of a well-defined glassy basin, from which the system does not escape, is thus relevant. The nature of the glassy basin, either smooth, rough, etc., determines the physical properties of the material. In this work, we study how the properties of glassy basins are affected by varying density.

More precisely, our strategy is to prepare equilibrium configurations at various state points $(\varphi_g, T_g)$ using hybrid swap, in order to select a glass basin, which we then follow out of equilibrium using ordinary MD simulations across the phase diagram $(\varphi, T)$. We determine how the properties of a given glass depend on the preparation state $(\varphi_g, T_g)$, which encodes its degree of stability, and the state point $(\varphi, T)$ at which it is studied. The glass properties depend both on initial and final state points. To each initial glass selected in the plane of Fig. 1, corresponds a two-dimensional state following phase diagram[17,37]. This makes a representation of the complete phase diagram difficult. Instead, we focus on a few well-chosen initial glasses, and follow their evolution with both $\varphi$ and $T$.

In order to connect the glassy physics to that of the jamming transition occurring at zero temperature, we use a conjugate gradient method to minimize all equilibrated state points in Fig. 1 to their inherent structure (IS)[5,6]. The potential being purely repulsive with a finite interaction range, the potential energy of the ISs can be either positive (indicating that some particles overlap), or zero (no overlap). We call "jammed" the former and

"unjammed" the latter ISs, and the jamming transition separates the two regimes[2]. In Fig. 1, open (full) circles reach unjammed (jammed) ISs under minimization. Depending on the preparation of the glass basin, in our model the jamming transition can occur over a range of packing fractions $\varphi_J \in [0.78 - 0.84]$. The lower value is found when minimizing random configurations, while the higher bound corresponds to the jamming transition of deeply thermalized samples. Note that the unusually high values of $\varphi_J$ (for a $3d$ system) stem from both polydispersity and non-additivite interactions of our mixture (see the Methods section).

**Ergodicity breaking inside a glassy minimum**. Configurations prepared by hybrid swap at a state point $(\varphi_g, T_g)$ are equilibrated, but their dynamics, when simulated by MD, is arrested, and no diffusion is observed. Yet, at long times, MD ergodically samples the glass basin selected by the initial configuration. We now study how this ergodicity is broken when temperature and packing fraction are changed. Let us stress that we are looking at an ergodicity breaking transition inside the glass, i.e. within the vibrational dynamics[30,31], which is very different from the more familiar ergodicity breaking transition occurring in the diffusive dynamics when quickly cooling the liquid into the glass phase.

Ergodicity breaking inside glassy minima can be detected using the isoconfigurational ensemble[41]. We prepare $n_c$ identical clones of each equilibrium configuration, initialized with independent velocities. Their dynamical evolution is studied at a new state point $(\varphi, T)$ with MD. We measure $\Delta_{AB}(t_w)$, the mean-squared displacement (MSD) between two clones after a time $t_w$ spent at the new state point, and $\Delta(t_w, t_w + \tau)$, the MSD of particles between time $t_w$ and $t_w + \tau$, in a single clone. Both quantities are averaged over clones and initial glasses (see the Methods section for details). The full time and waiting-time dependence of these quantities is studied below. Here we focus on their long time behavior. In the temperature regime studied, diffusion is suppressed and the MSDs typically show a plateau at long times, which we characterize by their value at $t_w = 8192$, $\tau = 10^4$. If the glass basin is sampled ergodically, the average distance between two clones must coincide with the average displacement performed by one clone, and $\Delta$ and $\Delta_{AB}$ should coincide[31]. Ergodicity breaking inside the glass is indicated by a strict inequality $\Delta < \Delta_{AB}$ in the long time limit.

We first consider the glasses prepared at $(\varphi_g, T_g) = (0.84, 0.029)$, $(0.85, 0.0353)$, $(0.87, 0.053)$, $(0.9, 0.09)$, $(0.95, 0.16)$, $(1, 0.26)$ shown in Fig. 1. These glasses are subjected to constant-density temperature quenches. The long-time limits of $\Delta$ and $\Delta_{AB}$ after the quenches are presented in Fig. 2. For all glasses, $\Delta = \Delta_{AB}$ around $T_g$ and slightly below. At lower temperatures however, we find that $\Delta_{AB}$ is systematically greater than $\Delta$. This means that the particles in different clones are further apart than what thermal fluctuations allow them to explore. The different clones become confined in distinct minima, which are dynamically inaccessible at low temperature: ergodicity is lost inside the glass. This ergodicity breaking transition inside the glass is conceptually different from the usual ergodicity breaking transition observed as the fluid transforms into a glass. The latter corresponds to the freezing of translational degrees of freedom while the phenomena discussed here corresponds to the freezing of vibrational degrees of freedom[31].

To better characterize this loss of ergodicity, we empirically define the temperature $T_G$ at which the two MSD separate as $\Delta_{AB} = 1.06\Delta$ (both values taken at $t_w = 8192$, $\tau = 10^4$). The arrows in Fig. 2 indicate the value $T_G$ for each glass studied.

We can also follow glasses both in temperature and packing fraction. We focus on two extreme regimes: $(\varphi_g, T_g) = (0.85, 0.0353)$ and $(1, 0.26)$ (see Fig. 1). The glasses are brought

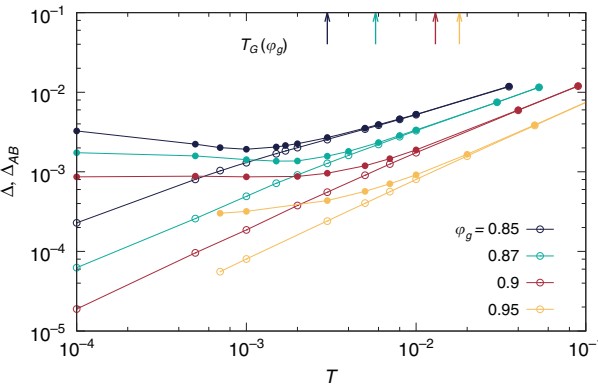

**Fig. 2** Loss of ergodicity at low temperature. Glasses prepared at various $(\varphi_g, T_g)$ and comparable stabilities are quenched to $T$ at fixed packing fraction. The long-time limit ($t_w = 8192$, $\tau = 10^4$) of the mean-squared displacement $\Delta$ (open symbols) and mean-squared distance $\Delta_{AB}$ (closed symbols) are shown as a function of the quench temperature. For each $\varphi_g$, a vertical arrow indicates the temperature $T_G$ at which the two MSD separate, signaling loss of ergodicity inside the glass basin

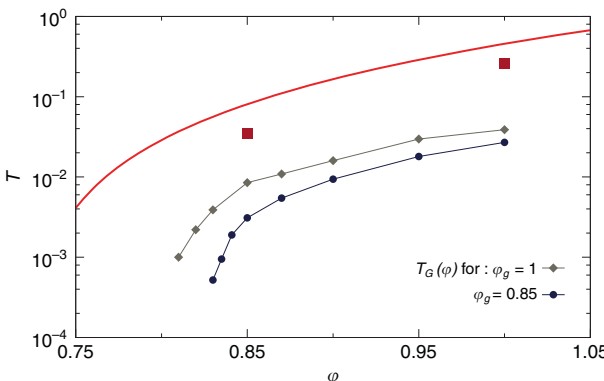

**Fig. 3** Loss of ergodicity for various glasses. Glasses are prepared in equilibrium conditions with the swap method at $(\varphi_g, T_g) = (0.85, 0.0353)$ and $(1, 0.26)$ (red squares), below the computer glass transition (red line). The glasses are rapidly brought to another state point $(\varphi, T)$ in the phase diagram. We compare the long-time limit of the mean-squared displacement and the mean-squared distance between clones of the glass at the new state point. Both values are equal around $(\varphi_g, T_g)$ where the glass basin is sampled ergodically. Ergodicity inside the glass is broken below $T_G(\varphi)$. The ergodicity breaking line $T_G(\varphi)$ depends on glass preparation, encoded in $(\varphi_g, T_g)$. We find that the lines $T_G(\varphi)$ for different glass preparations have the same qualitative behavior

instantaneously to a new state point $(\varphi, T)$, where the long-time limit of the MSDs is measured. Using the same criterion as above, we now look for the line $T_G(\varphi)$ at which ergodicity is broken. We report in Fig. 3 the $T_G$ line for the two initial glasses. Despite differences in the protocols, all the resulting ergodicity breaking temperatures $T_G(\varphi)$ behave qualitatively similarly and grow monotonically from zero above $\varphi \approx 0.81$. Hence, ergodicity within a glass basin is highly sensitive to the final state $(\varphi, T)$ to which the glass is brought, and the preparation history of the glass seems to simply set the overall scale of the barriers inside the glass. Consequently, we can use isochoric temperature quenches or compression/cooling protocols interchangeably.

The phase diagram in Fig. 3 suggests the following picture. At temperatures slightly below the preparation state $(\varphi_g, T_g)$, thermal fluctuations enable an ergodic sampling of the restricted

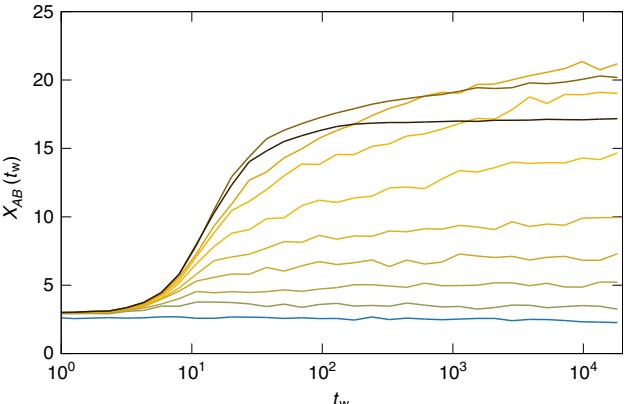

**Fig. 4** Growing susceptibility after a temperature quench. Glasses prepared at $(\varphi_g, T_g) = (0.85, 0.0353)$ are rapidly quenched to temperature $T$. From bottom to top, $T = 0.0353$, 0.005, 0.003, 0.002, 0.0015, 0.001, $7 \times 10^{-4}$, $5 \times 10^{-4}$, $10^{-4}$, $2 \times 10^{-5}$. The susceptibility $\chi_{AB}$, which quantifies the number of particles moving collectively, increases with time after quenches at $T < 0.002$. The value at long time $\chi_{AB}(t_w \simeq 10^4)$ increases with decreasing $T$, until a reverse of trend at $T < 5 \times 10^{-4}$. The non-monotonicity stems from the competition between the emergence of an increasingly complex landscape, which increases the susceptibility, and the dynamic slowdown due to the reduction of thermal fluctuations, which makes the exploration of that landscape more difficult at small temperature

portion of phase space defining a glass basin. When temperature is lowered, the clones retain the same particle arrangement, defined by the initial equilibrium configuration, but may fall into different sub-basins. The sub-basins may differ by the positions of a few particles, or the whole system, as discussed extensively below. The barrier crossing between the different sub-basins is associated to a temperature-dependent timescale. Ergodicity is lost when this timescale becomes much larger than the simulation time and clones then explore separate regions of phase space.

**Collective and heterogeneous dynamics**. In order to reveal the mechanism behind ergodicity breaking and the corresponding growing timescale, we investigate the existence of a growing lengthscale associated to microscopic vibrational dynamics. We use a dynamical susceptibility $\chi_{AB}(t_w)$, which provides an estimate of the number of correlated particles in the vibrational dynamics at time $t_w$. Its definition, given in the Methods section, ensures $\chi_{AB} \simeq 1$ for uncorrelated dynamics.

We present in Fig. 4 the time evolution of $\chi_{AB}(t_w)$ after quenching a glass prepared at $(\varphi_g, T_g) = (0.85, 0.0353)$ down to various temperatures $T$. For quenches slightly below $T_g$, the value of $\chi_{AB}$ remains of order unity at all times. The dynamics is ergodic and spatially uncorrelated. The susceptibility increases gradually in time and in amplitude with decreasing the target temperature. For quenches at very low temperature, the initial growth of the susceptibility with time is abrupt, but slows down dramatically at larger times. The value $\chi_{AB}(t_w = 10^4)$ is thus non-monotonic with $T$. While this behavior resembles qualitatively that reported in hard sphere glasses[31,32] and $3d$ spin glass models in an external field[42], to our knowledge, this is the first numerical evidence for a growing correlation lengthscale associated with vibrational dynamics in a model relevant to describe bulk thermal systems with soft interactions. The non-monotonicity can be interpreted as a competition between the emergence of an increasingly complex landscape, which tends to increase the susceptibility, and the dynamic slowdown due to the reduction of thermal fluctuations, which makes the exploration of that landscape more difficult at small $T$.

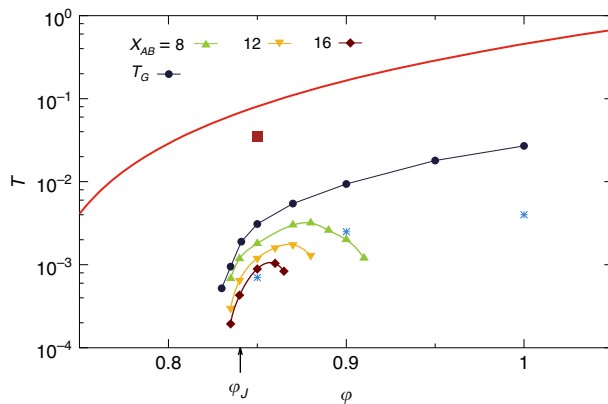

**Fig. 5** Ergodicity breaking versus collective dynamics. Glasses prepared at $(\varphi_g, T_g) = (0.85, 0.0353)$ (red square), below the computer glass transition (red line), are followed in the $\varphi - T$ phase diagram. We indicate the iso-$\chi_{AB}(t_w = 10^4)$ lines at which the dynamical susceptibility reaches 8, 12, and 16. The dynamics is increasingly collective in the dome delineated by theses lines. Ergodicity breaking, observed at the line $T_G(\varphi)$, may be due to collective defects (low $\varphi$), or not (high $\varphi$). The jamming transition $\varphi_J$ of the glass, indicated by an arrow, takes place under the dome where dynamics is collective. We show in Fig. 6 the microscopic dynamics after bringing the glass (red square) to three state points (blue stars) at which ergodicity is lost, and the dynamics is more or less collective (labeled a, b, c from left to right)

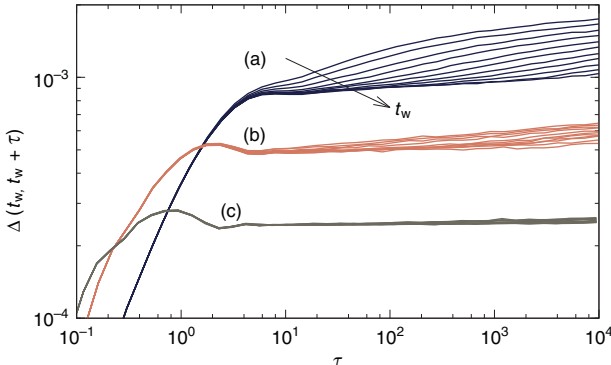

**Fig. 6** Microscopic dynamics after quenching a glass. Mean-squared displacement of glasses prepared in equilibrium at $(\varphi_g, T_g) = (0.85, 0.0353)$ then rapidly quenched to **a** $\varphi = 0.85$, $T = 7 \times 10^{-4}$; **b** $\varphi = 0.9$, $T = 0.0025$; **c** $\varphi = 1$, $T = 0.004$. For each set of curves, the time $t_w$ spent at low temperature increases from top to bottom: $t_w = 8, 16, 32, ..., 4096, 8192$. The three state points **a–c** are shown in Fig. 5 (blue stars). Strong aging effects are observed in **a**, mild aging in **b**, and no aging in **c**. The aging effect evidenced here takes place after further cooling dynamically arrested glasses. This effect is different from the common aging effect observed after rapidly cooling a liquid into the glass phase

We now determine the extent of the region in which the physics is governed by a complex landscape and spatially correlated dynamics. As in the Ergodicity breaking inside a glassy minimum section, we follow glasses initially prepared at $(\varphi_g, T_g) = (0.85, 0.0353)$ to state points $(\varphi, T)$, at which we measure the susceptibility $\chi_{AB}$, in particular its long-time value $\chi_{AB}(t_w = 10^4)$. In Fig. 5, we present iso-$\chi_{AB}(t_w = 10^4)$ lines which connect the state points at which the susceptibility reaches a value of 8, 12, or 16. The lines have a similar 'dome' shape, and delimit a region of the phase diagram which shrinks, both in temperature and packing fraction, as $\chi_{AB}$ increases. In this region of the phase diagram, the vibrational dynamics of the glass is slow, correlated in space, and heterogeneous. In the From extended defects near jamming to localized in dense liquids section, we characterize the structure of glassy minima in this region, and confirm that it has a highly complex organization.

We locate the jamming transition $(\varphi_J, T = 0)$ of the same glasses by slow decompression, followed by a gradual cooling of the system to $T = 0$. We find $\varphi_J \simeq 0.8404(5)$, indicated by an arrow in Fig. 5. Interestingly, the jamming transition is located inside the dome where the dynamics is governed by a complex landscape. We find a qualitatively similar behavior for glasses prepared at $\varphi_g = 1$, $T_g = 0.26$. As a result of our extensive exploration of all glassy regimes, we find that the emergence of an incredibly complex landscape at finite temperature, revealed by a growth of the dynamical susceptibility, always takes place close to the jamming transition in the phase diagram. Both phenomena are however distinct, since the 'dome' delimited by the iso-$\chi_{AB}$ lines extends up to temperature orders of magnitude larger than that of jamming criticality[43].

An important observation is that the loss of ergodicity and the increase of lengthscale do not coincide for all densities. In particular, for $\varphi \gtrsim 0.875$ the line $T_G(\varphi)$ is located at higher temperature than the iso-$\chi_{AB}$ lines. We conclude that the loss of ergodicity may have a different origin in different regions of the phase diagram. At lower densities, the loss of ergodicity is accompanied by a growth of $\chi_{AB}$, i.e. increasingly collective excitations. At higher packing fraction, the loss of ergodicity is not accompanied by a growing lengthscale, suggesting that it stems from localized defects. These results show that marginal stability, and its associated extended excitations, is not restricted to hard spheres, but can also be found for soft thermal systems in a region around jamming, which is our first important result.

**Time-evolution of the MSD**. We investigate the time-evolution of the MSD $\Delta$, and show that the vibrational dynamics becomes increasingly slow as it becomes more collective. We examine the dynamics of the glass at state points below the ergodicity breaking line $T_G(\varphi)$. We concentrate on three state points indicated by blue stars in Fig. 5. At these points, $\Delta_{AB}/\Delta$ takes a similar value, making their comparison meaningful.

We show in Fig. 6 the time evolution of the MSD $\Delta$ as a function of the waiting time $t_w$ after the quench. We observe very different behavior depending on the target state point. The microscopic dynamics exhibits a strong waiting-time dependence after a quench into the 'dome' (a). This represents a novel type of aging, different from more mundane aging effects observed after rapidly cooling a liquid below the glass transition. In our work, the glass is first equilibrated below the computer glass transition (which kills all diffusive processes), before being quenched to a lower temperature. The aging effect observed in Fig. 6 are related to the vibrational dynamics within a glass basin, whose bottom is so rough that relaxation processes involved after the quench are non-trivial, and have not reached steady state after a time $t_w = 8192$. Quenches further away from the dome, to $\varphi = 0.9$, $T = 0.0025$ (b), at which $\chi_{AB} \simeq 4$, lead to the same effects, but the amplitude of the decrease of $\Delta$ with waiting time is diminished compared to (a). Compressions to higher density $\varphi = 1$, $T = 0.004$ (c), where the dynamics is not collective, lead to stationary dynamics. This suggests that the glassy basin has a simpler structure at this state point. We explored compressions to $\varphi = 1$ and temperatures ranging from $T_G$ to very low temperature, and never observed any aging effects. We conclude that aging dynamics is a direct signature of the collective excitations taking place inside the glass[44]. We will confirm this picture in Figs. 7 and 8.

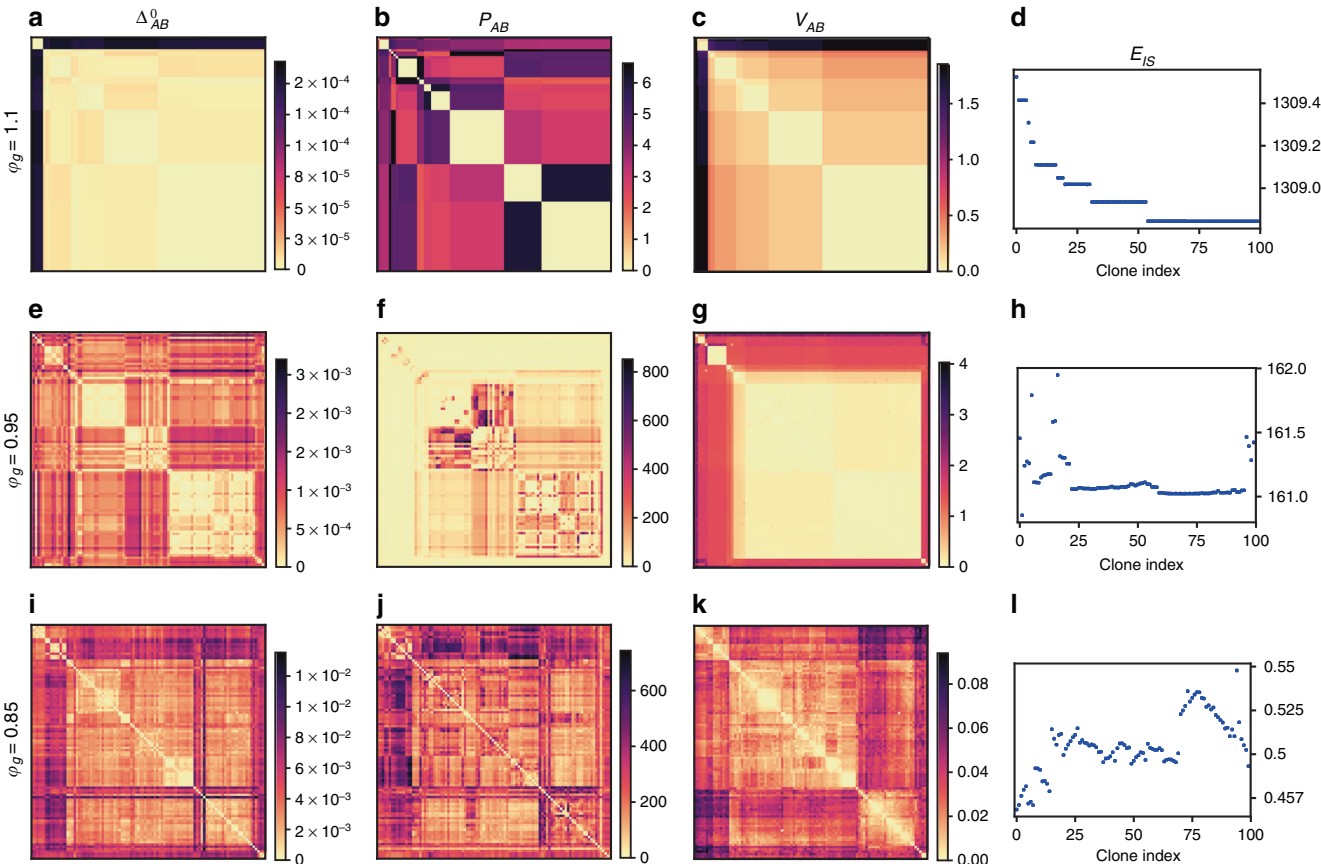

**Fig. 7** Crossover from simple to hierarchical landscapes in glasses prepared at $\varphi_g = 1.1$ (top row), 0.95 (middle), 0.85 (bottom). The potential energy landscape of glasses is sampled by $n_c = 100$ independent clones. We characterize the landscape by several observables $O$, presented as matrices $O_{AB}$, where $A$ and $B$ denote different clones, minimized to their inherent structure. We show the real-space distance $\Delta^0_{AB}$ **a**, **e**, **i**, participation ratio $P_{AB}$ **b**, **f**, **j**, and potential energy barrier $V_{AB}$ **c**, **g**, **k** between all pairs of clones. The potential energy $E_{IS}$ of the clones in their inherent structure is presented in **d**, **h**, **l**. The clones indexes are ordered by running a hierarchical clustering algorithm on the matrix $V_{AB}$. For each glass preparation $\varphi_g$, all observables are presented with the same clone ordering

**From extended defects near jamming to localized in dense liquids**. The phase diagram in Fig. 5, together with the aging results in the Time-evolution of the mean-squared displacement section, suggest the following picture. At low densities, $\varphi \sim 0.85$, the relaxation processes responsible for the loss of ergodicity are collective and extended, which naturally explains the growth of $\chi_{AB}$ and collective aging dynamics observed at low temperature. At high densities, $\varphi \sim 1.1$, these processes are localized and do not give rise to an increasing $\chi_{AB}$ or to aging dynamics. The crossover between these two extremes takes place around $\varphi \sim 0.95$ by a mechanism in which the barriers associated to extended defects are pushed towards lower energies, while localized defects with higher energy barriers appear. Ergodicity breaking and susceptibility growth are observed on these different temperature scales, since the former is caused by the localized defects, and the latter by extended ones.

To confirm this physical picture, we analyze the energy landscape of glasses prepared at $(\varphi_g, T_g) = (0.85, 0.0353)$, $(0.95, 0.16)$, and $(1.1, 0.46)$. For each state point, we select randomly one equilibrium configuration, which defines a glassy basin. In order to characterize its structure, we run simulations in the isoconfigurational ensemble. We create $n_c = 100$ clones of each initial equilibrium configuration. The clones are cooled to $T = 0.0005$, 0.005, and 0.0005, respectively. Our results do not depend on this choice of temperature. After a simulation time of $t_w = 10^4$, each clone is brought to its IS, where its potential energy $E_{IS}$ is measured. Different clones may or may not end up in the same IS, depending on the complexity of the landscape. In each glass, we analyze all pairs

$AB$ of clones after minimization. We compute: (i) the mean-squared distance $\Delta^0_{AB}$ between clones in their IS, (ii) the participation ratio $P_{AB}$ which indicates how many particles dominate the displacement field between the two IS, and (iii) the energy barrier $V_{AB}$ between the two IS (see the Time-evolution of the MSD section for technical details). To the best of our knowledge, such a complete study of the distribution of minima and barriers inside a glass basin has not been previously reported in the literature.

Representative results for the four observables ($\Delta^0_{AB}$, $P_{AB}$, $V_{AB}$, $E_{IS}$), and three glasses are presented in Fig. 7. The observables defined for pairs of clones are presented as square matrices. The clones are reordered to reveal a possible hierarchical structure of the landscape. In practice, this clustering is performed on the matrix of energy barriers $V_{AB}$ (see the Methods section) and is kept identical for all observables. The clustering allows to visualize easily the topology of the glass energy landscape (Fig. 7c, g, and k), and to follow its evolution with the glass preparation density.

We first describe Fig. 7a–d, which correspond to the glass prepared at high density $\varphi_g = 1.1$. We identify mainly two clusters of clones in Fig. 7a, c: one cluster contains only a few clones (top left), and the other contains the majority of clones. The clones inside a single cluster are close to one another (Fig. 7a), have similar energies (Fig. 7d), and are separated by low energy barriers (Fig. 7c). The two clusters are separated by a large energy barrier (Fig. 7c). The glass basin has little structure: it consists in two sub-basins separated by a high barrier. Each sub-basin contains a small number of IS (Fig. 7d). Whereas the

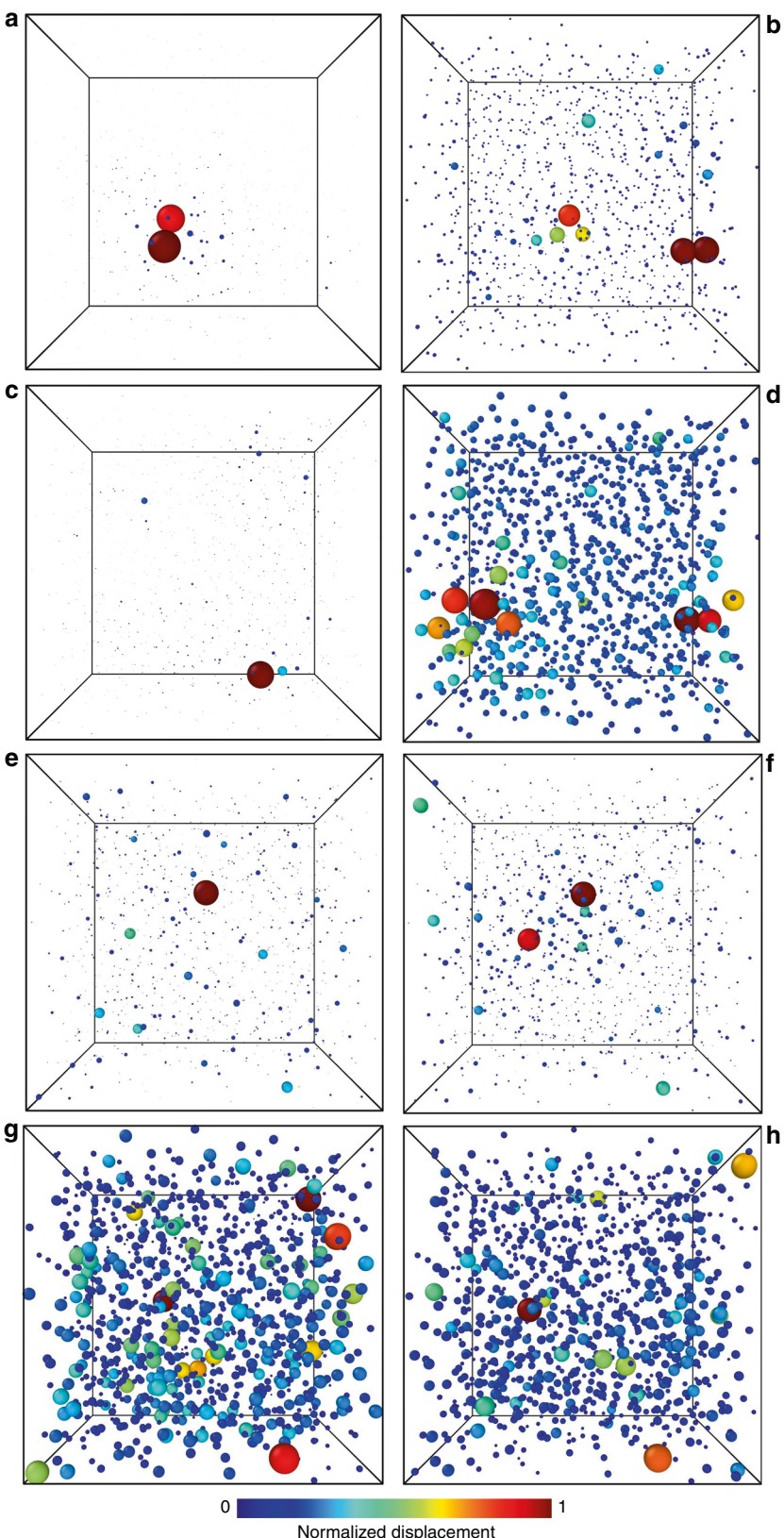

**Fig. 8** Coexistence of localized and extended defects in glasses. Snapshots of particle displacements between pairs of inherent structures of the same glass for **a** $\varphi_g = 1.1$; **b**, **c**, **d** $\varphi_g = 0.95$; **e**, **f**, **g**, **h** $\varphi_g = 0.85$. **a** $\varphi_g = 1.1$: localized defect, high-energy barrier; $\varphi_g = 0.95$: **b** localized, small barrier, **c** localized, high barrier, **d** delocalized, low barrier; $\varphi_g = 0.85$: **e** localized, small barrier; **f** localized, high barrier; **g** delocalized, small barrier; **h** delocalized, high barrier. Particle size and color are proportional to the particle displacement, normalized to the largest displacement in the sample

distance between the sub-basins is relatively large (Fig. 7a), they differ only by the position of a few particles, as shown by the maximum participation ratio $P_{AB} = 6$ (Fig. 7b). The defects inside this glass are simple: a few particles hop from one sub-basin to another. These localized defects control the separation of $\Delta_{AB}$ and $\Delta$ which arises at a temperature $T_G$ set by the large barrier $V_{AB}$ between the two clusters. Because they are simple and highly localized, these defects cannot give rise to aging dynamics (Fig. 6), or a growing susceptibility $\chi_{AB}$. Localized defects are crucial to understand the low-temperature thermal properties of glasses[7,8,22]. In light of the remarkable ability of the swap algorithm to create glasses with quench rates comparable to those found in vapor deposition studies, it would be interesting to analyze how the classical and quantum properties of these localized defects depend on glass preparation[45].

We then analyze in Fig. 7i–l the other extreme of a glass prepared at $\varphi_g = 0.85$. In this case, the landscape is instead very rough and extremely complex. The glass basin is characterized by a large number of distinct minima (Fig. 7l), separated by barriers of all sizes (Fig. 7k). The clustering in Fig. 7k suggests that the landscape is organized hierarchically. Contrary to the glass at $\varphi_g = 1.1$, there is little correlation between the energy barrier separating two minima (Fig. 7k) and the distance between them (Fig. 7i). There is however a good correlation between the distance (Fig. 7i) and participation ratio (Fig. 7j) matrices: extended excitations typically correspond to larger displacements. In summary, the glassy minimum contains barriers of all sizes and of all nature, from localized to extended. This explains why in this glass, the separation of $\Delta_{AB}$ and $\Delta$ around $T_G$ is concomitant with the growth of the susceptibility (Fig. 5) and the emergence of rejuvenation effects (Fig. 6a). To the best of our knowledge, this is the first time that such effects are reported in a model for thermal soft particles, relevant for dense colloidal suspensions, and emulsions.

We complete the above picture by providing a microscopic description of how defects transform from localized to extended as packing fraction decreases. Several scenarios could explain this transformation. The density of defects could remain the same, but individual defects become gradually extended. Alternatively, more and more localized defects could appear, and eventually percolate to form extended defects. We show in Fig. 7e–h that both types of defects and excitations coexist in the intermediate regime of densities, $\varphi_g = 0.95$. As for $\varphi_g = 1.1$, the glassy minimum is separated into a few sub-minima separated by large barriers. The barriers inside each sub-minimum are much smaller (Fig. 7g, h). Yet, the number of distinct minima inside each cluster is quite large (Fig. 7h). Strikingly, while the large-energy barriers (Fig. 7g) have a low participation ratio (Fig. 7f), the barriers found inside the sub-basins can be very collective, as for $\varphi_g = 0.85$. The landscape is clearly organized around two main energy scales: the scale $T_G$, at which ergodicity is lost due to the highest energy barriers associated to localized defects, which explains why neither a significant growth in the susceptibility, nor aging dynamics are observed around this temperature scale; and a lower scale, associated to extended defects, which controls the growth of $\chi_{AB}$ and the emergence of aging, Fig. 6b. We believe that this coexistence of extended and localized excitations, which is our second important result, will be manifested in the physical properties of the corresponding glasses, e.g. in transport properties[8], in the response to localized probes[7], and in the aging/rejuvenation dynamics after a sudden temperature change[44].

We accompany these conclusions with real-space snapshots illustrating the difference between distinct ISs in Fig. 8 resolved at the particle scale. For $\varphi_g = 1.1$, we show the localized defect responsible for the loss of ergodicity (Fig. 8a). Two particles move between two local minima, and the surrounding particles move

slightly due to the elasticity of the solid. For $\varphi_g = 0.95$, localized excitations can correspond to a low (Fig. 8b) or high (Fig. 8c) energy barrier. Delocalized excitations only correspond to the bottom of the landscape, i.e. to small barriers (Fig. 8d). For $\varphi_g = 0.85$, we find localized (Fig. 8e, f) and delocalized (Fig. 8g, h) excitations, which are either associated to small barriers (Fig. 8e, g) or high-energy barriers (Fig. 8f, h) demonstrating the variety and increasing complexity of the potential energy landscape when moving closer to the jamming transition.

## Discussion

We study the nature of excitations and defects through extensive simulations of a three-dimensional WCA glass former. Our main finding is that the nature of the energy landscape can be sensitively tuned by changing density. At high densities, in the regime relevant for molecular and atomic glasses, the landscape is rather simple, characterized by few minima. The dynamical behavior of the glass is dictated by highly localized defects, which correspond to a few particles hopping between nearby configurations. This corresponds to the standard picture of two-level systems in glasses[7,22]. By contrast, at lower densities, relevant for granular materials, soft and hard colloidal suspensions, the landscape is very rough and has a hierarchical structure. There exist barriers over a broad range of energy scales, with a degree of localization that spans very localized and highly extended defects. This first result extends to finite temperatures earlier results about the marginality associated to athermal jamming[2,25,28,29]. Most interestingly, in the intermediate regime of densities, relevant for soft colloidal particles and emulsions, the landscape is characterized by both features. Localized defects dominate at higher temperature, and are responsible for ergodicity breaking inside the glass. The freezing of these defects, which involve a few particles, defines a small number of sub-basins. Each sub-basin, however, possesses a complex structure at lower energy scales, with extended defects associated to low barriers that appear similar to the ones found at lower densities. This second result leads us to predict that soft colloidal glasses and emulsions should be characterized by a complex hierarchical landscape, giving rise to interesting new physics, such as ergodicity breaking transitions, aging in the glass, rejuvenation and memory effects[44].

We showed that in all physical regimes, low-temperature glasses evolve inside an energy basin composed of a potentially large number of sub-minima. This sub-structure gives rise to a new type of ergodicity breaking transition in glasses at low temperature. The ergodicity breaking transition may or may not be accompanied by a growing lengthscale, dynamic heterogeneity and aging effects, depending on the preparation density of the glass. One can thus expect a variety of behaviors in distinct materials, depending on their location in the phase diagram analyzed in our work. Our study paves the way to a complete numerical and experimental characterization of the low-temperature behavior of glasses, including in the quantum regime where tunneling properties are likely to be strongly influenced by the spatial nature of defects.

## Methods
**Model**. We study a three-dimensional WCA model of soft repulsive particles[36]. The pair interaction between particles $i$ and $j$ reads

$$V = \begin{cases} 4\epsilon \left[ \left( \frac{\sigma_{ij}}{r_{ij}} \right)^{12} - \left( \frac{\sigma_{ij}}{r_{ij}} \right)^{6} \right] + 1, & r_{ij} < 2^{1/6}\sigma_{ij} \\ 0 & \text{otherwise.} \end{cases} \quad (1)$$

The potential and its first derivative are smooth at the cutoff distance $\bar{\sigma}_{ij} = 2^{1/6}\sigma_{ij}$. We use a non-additive polydisperse mixture to stabilize the homogeneous fluid against fractionation or crystallization[40], $\sigma_{ij} = \frac{1}{2}(\sigma_i + \sigma_j)\left(1 - 0.2|\sigma_i - \sigma_j|\right)$. Although the potential is non-additive, the quantity $\bar{\sigma}_i = 2^{1/6}\sigma_i$ acts as an effective particle diameter. The $\sigma_i$ are distributed continuously with the distribution

$P(\sigma_m \leq \sigma \leq \sigma_M) \sim 1/\sigma^3$, with a size ratio $\sigma_m/\sigma_M = 0.45$, and $\langle\sigma\rangle = \int P(\sigma)d\sigma = 1$. The polydispersity of the system is 23%. We study systems of $N = 1000$ particles of mass $m$, in a box of linear size $L$ and volume $V = L^3$. The relevant control parameters are the temperature $T$ and the packing fraction $\varphi$, defined as $\varphi = \pi/(6V)\sum_i \bar{\sigma}_i^3 = \sqrt{2}\pi/(6V)\sum_i \sigma_i^3$. The effect of polydispersity and non-additivity is to shift all packing fractions to higher values. For example the jamming transition of random packings is equal to $\varphi_J \simeq 0.78$ in our model, instead of 0.64 for monodisperse $3d$ packings with additive interactions. Energies, lengths and times are, respectively, expressed in units of $\epsilon$, $\sqrt{\epsilon/m\langle\sigma\rangle^2}$, and $\langle\sigma\rangle$.

**Preparation protocols**. We employ a two-step protocol. First, we generate equilibrated configurations at various state points $(\varphi_g, T_g)$, for which the physical dynamics is completely arrested on accessible time scales (see Fig. 1) but where thermalization can be achieved using a hybrid swap Monte Carlo technique[40]. For the hybrid swap, we use the implementation of ref. [38]. For each state point $(\varphi_g, T_g)$, we first generate $n_g = 200$ independent equilibrium configurations. Second, we use these very stable equilibrium configurations as input for MD simulations. The equations of motion are solved with an integration timestep $dt = 0.0035$. The temperature of the system is imposed by a Berendsen thermostat with a timescale $\tau_B = 1.0$. We consider thermodynamic conditions where particle diffusion is fully arrested, so that simulations always remain confined within a single glass basin, selected by the initial configuration. In addition, we produce $n_c = 20$ clones for each of the $n_g$ initial configurations (we use $n_c = 100$ for Fig. 7). Clones share the same initial positions, but are given initial velocities randomly sampled from the appropriate Maxwell distribution. Each configuration is then studied at various $(\varphi, T)$ by instantaneously changing the control parameters. This two-step process emulates the 'state following' construction employed in mean-field analytical studies[46,47], that we have recently applied to the WCA model[37]. This protocol is also a fair numerical implementation of an experimental protocol where glasses are produced by cooling, and the glassy state frozen at the experimental glass transition temperature is then studied at various state points within the glass phase.

**Observables for ergodicity breaking in the glass basin**. From a given initial equilibrated configuration at $(\varphi_g, T_g)$, we run $n_c$ independent MD simulations using the clones as initial conditions. At the beginning of the simulation, the control parameters $(\varphi, T)$ are changed instantaneously. It takes a time $t \sim 10$ for the kinetic temperature to reach the desired value. The origin for waiting times $t_w = 0$ is defined as the time at which the kinetic temperature, averaged over a time $t = 1$, reaches the imposed value. We focus on two measures of distance: the MSD $\Delta_{AB}$ between the same particles in different clones of a glass,

$$\Delta_{AB}(t_w) = \frac{1}{N_b}\sum_{i=1}^{N_b}\langle|\mathbf{r}_i^A(t_w) - \mathbf{r}_i^B(t_w)|^2\rangle, \qquad (2)$$

and the MSD $\Delta$, which quantifies the dynamics of the particles within a single clone,

$$\Delta(t_w, t_w + \tau) = \frac{1}{N_b}\sum_{i=1}^{N_b}\langle|\mathbf{r}_i(t_w + \tau) - \mathbf{r}_i(t_w)|^2\rangle. \qquad (3)$$

Here, $\mathbf{r}_i$ is the coordinate of particle $i$, and $\mathbf{r}_i^A$ and $\mathbf{r}_i^B$ the positions of particle $i$ in two different clones, which are generically referred to as $A$ and $B$. The average is made using the $N_b = N/2$ particles having the largest diameter. We find that smaller particles are more mobile and sometimes dominate the average. The brackets indicate an average both on disorder (using the $n_g = 200$ independent initial configurations), and thermal history (using the $n_c = 20$ clones for each glass). In the case of $\Delta_{AB}$, the thermal average is performed over the $n_c(n_c - 1)/2$ pairs of clones.

We define the susceptibility associated to the global fluctuations of the mean-squared distance between clones[23,31],

$$\chi_{AB} = N_b \frac{\langle\Delta_{AB}^2\rangle - \langle\Delta_{AB}\rangle^2}{\langle\Delta_{AB}^{i\,2}\rangle - \langle\Delta_{AB}^i\rangle^2}, \qquad (4)$$

where $\Delta_{AB}$ is defined in Eq. (2), and $\Delta_{AB}^i$ represents its single-particle version. The time dependence in Eq. (4) is omitted to ease the reading, but just as $\Delta_{AB}$, $\chi_{AB}(t_w)$ is a time-dependent observable. The normalization in $\chi_{AB}$ ensures that $\chi_{AB} = 1$ for spatially uncorrelated dynamics. Using this definition, $\chi_{AB}$ is also a direct measure of a correlation volume, and it is the direct analog of a spin-glass susceptibility.

**Exploration of the potential energy landscape**. To explore the energy landscape associated to a given initial equilibrium configuration we first create $n_c = 100$ clones. The clones are cooled to a lower temperature: $T = 0.0005, 0.005$, and $0.0005$, for the glasses prepared at $\varphi_g = 1.1, 0.95, 0.85$, respectively. We simulate the dynamics of these systems at these low temperatures $T$ during a total time $t_w = 10^4$. At the end of the simulation, the configuration is minimized using a conjugate gradient algorithm to bring each clone to its IS. We measure the potential energy $E_{IS}$ of each IS found inside each glass basin.

We then compare all pairs of IS found inside each glass. We compute the distance between two IS, taking into account all $N$ particles:

$$\Delta_{AB}^0 = \frac{1}{N}\sum_i |\mathbf{r}_i^{A,0} - \mathbf{r}_i^{B,0}|^2, \qquad (5)$$

where $\mathbf{r}_i^{A,0}$ is the position of particle $i$ in the IS of clone $A$. In order to have a more refined information on how many particles contribute to the value of $\Delta_{AB}$, we compute a participation ratio

$$P_{AB} = \frac{\left[\sum_{\mu,i}(\delta r_{AB}^{\mu,i})^2\right]^2}{\sum_{\mu,i}(\delta r_{AB}^{\mu,i})^4}, \qquad (6)$$

where $\delta r_{AB}^{\mu,i} = \mu_i^{A,0} - \mu_i^{B,0}$ ($\mu = x, y, z$). The sum runs over all $N$ particles. With this definition, the participation ratio directly estimates the number of particles which dominate the difference between pairs of IS.

For each pair of IS, we estimate an energy barrier using the nudge elastic band (NEB) method[48]. Note that this approximate method only provides an upper bound for the lowest energy barrier separating the two energy minima. We use 40 intermediate images of the system, initialized by linear interpolation between two IS. We relax the chain of images using the potential energy Eq. (1) in directions transverse to the chain, and elastic springs in the parallel direction. We use a climbing version of the method[49], which ensures that one image is at the saddle point. The energy barrier $V_{AB}$ is the energy difference between the saddle point and the lowest energy minimum.

We find that for $\varphi \lesssim 0.9$, the NEB method does not converge properly. The reason is that at low temperature in this density regime, the particles behave more and more like hard sphere particles, for which the potential energy cost of overlapping particles diverges. In some cases, the linear interpolation creates strongly overlapping particles, and a singularity in the potential energy of the chain of images. To work around this problem, we first perform a NEB minimization using a harmonic repulsion between particles, instead of WCA. The harmonic potential used is $V = 18 \times 2^{2/3}(r_{ij}/\sigma_{ij} - 2^{1/6})^2$ if $r_{ij} < 2^{1/6}\sigma_{ij}$ and zero otherwise. The WCA and this harmonic potential have the same first two derivatives at the cutoff. Both potentials behave similarly at small overlaps, but the harmonic one does not create diverging potential energies due to particle overlaps. The initial NEB minimization run with harmonic repulsion converges smoothly at all densities, and removes spurious particle overlaps. The relaxed chain of images is then minimized with the NEB method using the original WCA potential. We have checked that both methods (WCA versus harmonic + WCA) yield similar results at high densities $\varphi_g = 1.1$ and $\varphi_g = 0.95$. This validates the two-step NEB minimization using the combination of harmonic and WCA interactions that we implement at $\varphi_g = 0.85$.

The hierarchical clustering is performed on the barrier matrix $V_{AB}$ using the `linkage` function of the 'hierarchical clustering' python package, which is an agglomerative algorithm[50]. Initially, each clone starts in its own cluster. The clusters are gradually merged until they form one large cluster. The merging rule is defined by giving: a distance between individual clones (here, the energy barrier $V_{AB}$), and a 'linkage criterion' which defines the distance between clusters. At each step, clusters with the smallest distance are merged. We find empirically that an 'average' linkage (the distance between clusters is the average of the energy barriers on all pairs) gives a good enough clustering.

The snapshots shown in Fig. 8 highlight the displacement of particles between two IS. The particle positions are those of one IS, and the size and color code for the particles is proportional to the displacement $|\mathbf{r}_i^{A,0} - \mathbf{r}_i^{B,0}|$ between the IS. The particle with largest displacement is set to a diameter 1. This allows to compare visually snapshots of systems for which the displacement may vary by orders of magnitude. The snapshots should therefore be read in parallel with Fig. 7a, e, and i, which provide the scale for particle displacements in each case.

## Data availability
The data that support the findings of this study are available upon request from the corresponding author.

## Code availability
The codes that support the findings of this study are available upon request from the corresponding author.

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

## Acknowledgements

We thank M. Baity-Jesi, W. Ji, D. Khomenko, B. Seoane, and D. Reichman for interesting discussions, and C. Dennis for hierarchical clustering tips. This project has received funding from the European Research Council (ERC) under the European Union's Horizon 2020 research and innovation program (grant agreement no. 723955—GlassUniversality). This work was supported by a grant from the Simons Foundation (#454933, L.B.; #454955, F.Z.).

## Author contributions

C.S., L.B., and F.Z. designed research. C.S. performed research and analyzed data. C.S., L.B., and F.Z. wrote the paper.

## Competing interests

The auhors declare no competing interests.
