## [Peer Review File · Nature Communications]

Reviewers' comments:

Reviewer #1 (Remarks to the Author):

Dear editor,

I have carefully reviewed manuscript # NCOMMS-19-18724 "Nature of excitations and defects in structural glasses," by C. Scalliet, L. Berthier, and F. Zamponi. The manuscript describes computational studies of slow dynamics of repulsive Lennard-Jones particles as a function of packing fraction and temperature. This manuscript is extremely well-written, the simulations and calculations are performed meticulously, and the main results, which describe the evolution of the dynamics from collective with rugged energy landscapes at low packing fractions near jamming onset to localized dynamics on more homogeneous energy landscapes, represent a major advance. There have been studies in the literature that hint at some of these results, but this work is a comprehensive study that clarifies the slow dynamics at low temperature as a function of packing fraction from the jamming regime to highly compressed systems.

Before giving my final recommendation, I would like the authors to address the following comments and questions.

1) The authors state in the introduction, "In the vicinity of the [jamming] transition, the solid is marginally stable [2,16]: a small perturbation can remove a few contacts and destabilize the entire solid." This point reminds me of studies in the literature about the breakdown of linear vibrational response of jammed solids at extremely small temperatures after "a few contacts" begin to break and the system moves to a new nearby minimum. See e.g. PRE 89, 062203 (2014) and PRE 96 (2017) 062902. It seems like this is relevant work and should be cited.

2) The difference between Lennard-Jones interactions, which model atomic glasses, and repulsive Lennard-Jones interactions are that Lennard-Jones potentials possess attractive interactions. One approach to compare these two classes of models is to treat the attractions as a small perturbation, but it is not clear whether this approach works when studying slow dynamics well below the putative glass transition. In the current work, the authors study a zeroth order approximation where there are no attractions.

Related to this, a number of studies have reported significant dynamical heterogeneities, growing correlation lengths, and aging in studies of Lennard-Jones systems (with attractions) near and below the glass transition. However, the current studies suggest that there is no aging, small correlation lengths, and smooth energy landscapes for overcompressed systems in the regime that is comparable to those used in Lennard-Jones systems.

Can the authors explain this possible contradiction? One possible remedy is

to show some of the same data for Lennard-Jones systems in Figs. 2-6.

3. The authors can remind the readers of the computer glass transition (red line) in the captions to Figs. 3 and 5.

4. The authors show the nice result that the zero-temperature jamming transition occurs at a packing fraction within the "dome" of strong collective dynamics in Fig. 5. As the authors know well, the packing fraction at jamming onset can occur over a wide range of packing fractions, from low values for non-thermalized protocols to extremely high values for thermalized protocols. What is the full range of jammed packing fractions for this system, when using a range of protocols from rapid energy minimization to annealing protocols? Are only a certain range of jammed packing fractions relevant for the current studies?

5. The authors find that zero-temperature jamming onset occurs near packing fraction $\phi_J \approx 0.840$. However, many weakly to moderately polydisperse systems jam at $\phi_J \approx 0.64$ in 3D. Is there an extra factor of $2^{1/6}$ in the definition of the packing fraction or do the small particles begin fitting in the interstices between larger particles for this polydispersity? If so, I would expect some differences in the dynamics of weakly polydisperse jammed systems and the current studies since the jamming region in weakly polydisperse systems is more than 30% lower than the current studies.

Looking forward to hearing your responses,

Corey O'Hern

Reviewer #2 (Remarks to the Author):

I read with certain interest the ms of Scalliet et al.

The authors have done a very complete numerical study of the WCA model using hybrid swaps and MD simulations to extrapolate (or interpolate) between the jamming and glass transitions.

The ms follows a series of well defined studies by the authors and collaborators.

In the present paper the authors tackle the topic of excitations and defects in such a system focusing on their effects for the two transitions separately and in unison: glass and jamming.

Technically, the paper is flawless.

However I cannot see the broad significance of the results.

The authors first study the equilibrium phase diagram of the model, then they proceed to study the loss of ergodicity and then they offer insight on this loss of ergodicity by studying the RMS displacement.

All these studies have been done in the past and have been investigated to exhaustion.

It is not clear from the ms what kind of new insight the authors bring to deserve a large interdisciplinary audience of Nat. Comm.

The problem is compounded by the fact that the authors give the impression that the topic is of

interest to a very narrow set of scientists.

Out of the 46 references, I counted 22 of the references given to either Zamponi, Biroli, Berthier or close collaborators of the group.

This is a self-defeating strategy to publish in a broad interdisciplinary journal, since the reader is left with the opinion that the topic is of interest to a narrow glassy community, which indeed, it might be the case, considering the main citations go only to the authors of the article.

The authors should make a bigger effort to expand on their view of the problem and show how the excitations in the glass and jamming transition are important for the rest of the community.

The entire purpose to publish in a Journal like Nature is to reach vast audiences beyond the close group of collaborators that already read this paper from the arXiv.

Indeed, at the end of each section, from ergodicity to RMS studies, the authors conclude that their simulations fairly agree with existing results, most of them from their own groups, so, the reader is left with the difficult task to figure out what is really new in the manuscript.

Having said that, I still believe that this is a complete study of excitations in glasses and if the authors can explain the broader significance of the results, it might have a chance to reach the Nature Comm audience.

They just need to make an effort to get out of their cage model of glassy dynamics and think more broadly than they are used to.

Reviewer #3 (Remarks to the Author):

Review attached.

This work by Scalliet, Berthier, and Zamponi provides a detailed characterization of the energy landscape of a 3D repulsive glass. This is an important contribution to our rapidly developing understanding of the glass transition that significantly extends previous work by the authors [Ref 27 in the current paper]. The authors find distinct regimes in the energy landscape of glasses that behave more like critically jammed solids with delocalized excitations, and these materials can be transformed into materials that behave akin to molecular and atomic glasses through changes in temperature and density. This is a deep dive into an important fundamental problem, and I think it should be published after the authors consider my comments below.

1. On page 4, right-hand column: the non-monotonicity in the susceptibility is interpreted as a growing length scale associated with the vibrational dynamics.
2. The conclusion on page 5 that the susceptibility growth is influenced by the underlying jamming transition seems quite vague. What is meant by this statement? Influenced in what way? What does this conclusion even mean other than trying to draw a connection between two phenomena that are nearby in phase space?

Minor points:

- n_c seems to be used with two different definitions: on page 5, section E, 2nd paragraph, I think it implies the number of independent configurations (different \mathbf{r}^N at the same thermodynamic condition), while on page 3, section B, second paragraph it seems to mean the number of replicates in the isoconfigurational ensemble used by the authors (same \mathbf{r}^N , same thermodynamic conditions, distinct velocity initializations).
- The citations are heavily biased towards self-citations. Things like swap MC have a long history of application, and the isoconfigurational ensemble approach used by the authors was developed by Harrowell and co-workers. The authors should at least credit the pioneers of these methods.

[...] *This manuscript is extremely well-written, the simulations and calculations are performed meticulously, and the main results, which describe the evolution of the dynamics from collective with rugged energy landscapes at low packing fractions near jamming onset to localized dynamics on more homogeneous energy landscapes, represent a major advance.*

We thank Reviewer 1 for the positive appreciation of our work and the constructive comments, which we address below. The corresponding changes made to the resubmitted manuscript are highlighted in red to facilitate its review.

1) *The authors state in the introduction, "In the vicinity of the [jamming] transition, the solid is marginally stable [2,16]: a small perturbation can remove a few contacts and destabilize the entire solid." This point reminds me of studies in the literature about the breakdown of linear vibrational response of jammed solids at extremely small temperatures after "a few contacts" begin to break and the system moves to a new nearby minimum. See e.g. PRE 89, 062203 (2014) and PRE 96 (2017) 062902. It seems like this is relevant work and should be cited.*

Reply: We thank the referee for pointing at these works. We added these references in the revised manuscript.

2) *The difference between Lennard-Jones interactions, which model atomic glasses, and repulsive Lennard-Jones interactions are that Lennard-Jones potentials possess attractive interactions. One approach to compare these two classes of models is to treat the attractions as a small perturbation, but it is not clear whether this approach works when studying slow dynamics well below the putative glass transition. In the current work, the authors study a zeroth order approximation where there are no attractions.*

Related to this, a number of studies have reported significant dynamical heterogeneities, growing correlation lengths, and aging in studies of Lennard-Jones systems (with attractions) near and below the glass transition. However, the current studies suggest that there is no aging, small correlation lengths, and smooth energy landscapes for overcompressed systems in the regime that is comparable to those used in Lennard-Jones systems.

Can the authors explain this possible contradiction? One possible remedy is to show some of the same data for Lennard-Jones systems in Figs. 2-6.

Reply: The referee is correct to point out that our interaction potential differs from the standard Lennard-Jones potential. However, our results are not in contradiction with those obtained in Lennard-Jones systems. Our WCA model glass-former *does* have dynamical heterogeneities and growing dynamic lengthscale in the supercooled liquid, and aging effects are observed after a sudden quench just below the glass transition [see e.g. Phys. Rev. E 86, 041506 (2012)].

The preparation protocol employed and temperature regime explored in our work is, however, entirely new. Our aim is not to study the aging of quickly quenched glasses (which has been characterized in depth in the past), but to study the excitations and defects of well-annealed (or even ultra-stable) glasses that display no physical aging on experimental time scales. We thus first prepare equilibrium configurations below the computer glass transition, then study their physical dynamics at even lower temperatures. The initial sample therefore corresponds to a very well annealed glass, whose aging dynamics would happen on timescales much larger than the accessible ones, and is therefore irrelevant. The fact that we do not observe aging upon further cooling of the glass does not contradict standard aging studies (rapid quench from the liquid) on Lennard-Jones glasses. The same phenomenology we observe would be observed in Lennard-Jones systems, if the same preparation protocol were used.

In the revised manuscript, we have made it clearer that the protocol and temperature regime in this work is different from previous numerical studies of ordinary aging dynamics.

3. The authors can remind the readers of the computer glass transition (red line) in the captions to Figs. 3 and 5.

Reply: We thank the referee for this suggestion. We added a sentence to both captions.

4. The authors show the nice result that the zero-temperature jamming transition occurs at a packing fraction within the "dome" of strong collective dynamics in Fig. 5. As the authors know well, the packing fraction at jamming onset can occur over a wide range of packing fractions, from low values for non-thermalized protocols to extremely high values for thermalized protocols. What is the full range of jammed packing fractions for this system, when using a range of protocols from rapid energy minimization to annealing protocols? Are only a certain range of jammed packing fractions relevant for the current studies?

Reply: The full range of jammed packing fraction we could observe in our study is $\phi_J \in [0.78 - 0.84]$ (which is quite a broad range—thanks to the swap algorithm). Despite this variety of jamming densities, for all initial glasses, prepared at various packing fraction (which therefore have a different ϕ_J), we find that there is a dome in temperature/density of collective dynamics around each jamming point.

The numerical protocol employed to detect the ‘dome’ requires to work with sufficiently well annealed glasses. If the glass is poorly annealed (corresponding to lower ϕ_J), its clones will rapidly drift apart in the course of the numerical simulation, and eventually fall into very different glassy minima. This represents however only a technical limitation rather than a conceptual difference.

5. The authors find that zero-temperature jamming onset occurs near packing fraction $\phi_J \approx 0.840$. However, many weakly to moderately polydisperse systems jam at $\phi_J \approx 0.64$ in 3D. Is there an extra factor of $2^{1/6}$ in the definition of the packing fraction or do the small

particles begin fitting in the interstices between larger particles for this polydispersity? If so, I would expect some differences in the dynamics of weakly polydisperse jammed systems and the current studies since the jamming region in weakly polydisperse systems is more than 30% lower than the current studies.

Reply: We understand that our values for jamming packing fraction appear unusually high for 3D systems. There are two reasons for this. The first reason is the degree of polydispersity of the system, which is equal to 23%. The second reason is the non-additive nature of interactions. We employ a non-additive rule for the cross-diameter $\sigma_{ij} = \frac{\sigma_i + \sigma_j}{2}(1 - 0.2|\sigma_i - \sigma_j|)$: two neighboring particles with significantly different diameters can come closer together than the standard $\frac{\sigma_i + \sigma_j}{2}$. Since we can only use the particle diameters in the definition of the packing fraction, the non-additive nature of interactions shifts in a non-trivial way all packing fractions to higher values compared to the well-known values. The well-known “0.64” value is not a directly relevant reference packing fraction for our system.

To be precise, for 3D hard-spheres with the same 23% polydispersity, and additive interactions, the range of jamming packing fraction is $\phi_J \in [0.655 - 0.71]$ (see Ozawa et al., SciPost Phys. 3, 027 (2017)). For the system under study (3D, 23% polydispersity, non additive interactions), we find $\phi_J \in [0.78 - 0.84]$. These numbers suggest indeed that non-additivity shifts the packing fraction values.

We have added the numbers for the range ϕ_J and provide an explanation for the high values in the manuscript.

The authors have done a very complete numerical study of the WCA model using hybrid swaps and MD simulations to extrapolate (or interpolate) between the jamming and glass transitions. The ms follows a series of well defined studies by the authors and collaborators. In the present paper the authors tackle the topic of excitations and defects in such a system focusing on their effects for the two transitions separately and in unison: glass and jamming.

Technically, the paper is flawless.

However I cannot see the broad significance of the results. The authors first study the equilibrium phase diagram of the model, then they proceed to study the loss of ergodicity and then they offer insight on this lose of ergodicity by studying the RMS displacement.

All these studies have been done in the past and have been investigated to exhaustion.

The problem is compounded by the fact that the authors give the impression that the topic is of interest to a very narrow set of scientists. Out the 46 references, I counted 22 of the references given to either Zamponi, Biroli, Berthier or close collaborators of the group. This is a self-defeating strategy to publish in a broad interdisciplinary journal, since the reader is left with the opinion that the topic is of interest to a narrow glassy community, which indeed, it might be the case, considering the main citations go only to the authors of the article.

The authors should make a bigger effort to expand on their view of the problem and show how the excitations in the glass and jamming transition are important for the rest of the community.

Indeed, at the end of each sections, from ergodicity to RMS studies, the authors conclude that their simulations fairly agree with existing results, most of them from their own groups, so, the reader is left with the difficult task to figure out what is really new in the manuscript.

Having said that, I still believe that this is a complete study of excitations in glasses and if the authors can explain the broader significance of the results, it might have a chance to reach the Nature Comm audience.

Reply:

We thank the referee for his/her critical reading of our manuscript, for recognizing its scientific interest, and for suggesting ways to improve its accessibility for a broad audience. The referee's main point is correct, and it is well taken.

First of all, we would like to stress once again the main elements of novelty that we introduce in this work with respect to previous works. From the technical point of view:

- We introduce a model in which a single physical control parameter, i.e. density, allows one to tune the energy landscape and interpolate continuously between the two previously-investigated extreme cases (hard and soft spheres).

- In addition to the “standard” RMS study, we perform a very detailed study of the energy landscape. We run the nudge elastic band (NEB) algorithm to identify the energy barriers between nearby minima (sec.II.E), which results in the “landscape maps” shown in Fig.7. To the best of our knowledge, such maps have not been shown before.

From the physical point of view:

- We show that the hard sphere physics of marginally stable glass with extended excitations is found in a whole “dome” around the jamming point, and it is thus relevant for soft spheres in a broad temperature-density range. We thus show that hard spheres are not an isolated, singular point in the interaction potential space.
- We continuously interpolate between the low density (hard-sphere-like with extended defects) and the high density (soft-sphere-like with localized defects) regimes, and discover a new intermediate regime characterized by a coexistence of extended and localized excitations, and a complex, hierarchically organized landscape, as shown in Fig.7.

We revised the whole manuscript to further highlight the novelty and broad significance of these results.

Second, we would like to apologize for having overlooked the high number of self-citations in our first submission. We wanted to situate our work in the context of recent previous work on Gardner-like physics, and this inadvertently led us to cite too many references of ours or close collaborators. We thus deeply revised the references, eliminated all redundant self-cites, and added several references to previous work of the broad community (keeping in mind that the total number of references is limited to 50).

[...] This is a deep dive into an important fundamental problem, and I think it should be published after the authors consider my comments below.

We thank Reviewer 3 for the positive review and recommendation to publish the manuscript at Nature Communications. Below, we address the Reviewer's comments and questions. The corresponding changes made to the resubmitted manuscript are highlighted in red to facilitate its review.

1. On page 4, right-hand column: the non-monotonicity in the susceptibility is interpreted as a growing length scale associated with the vibrational dynamics.

Reply: Yes, this is indeed the interpretation we had in mind. We are not sure to understand if the referee is requesting some more explanation of this point. We added more details on this connection.

2. The conclusion on page 5 that the susceptibility growth is influenced by the underlying jamming transition seems quite vague. What is meant by this statement? Influenced in what way? What does this conclusion even mean other than trying to draw a connection between two phenomena that are nearby in phase space?

Reply: We agree that this statement is too vague. The only thing that we are sure about is that a growing of the susceptibility is never observed far from jamming (at high packing fractions). We deduce from this observation that both phenomena take place nearby in parameter space. On the other hand, we agree that there is no evidence for a direct connection between the two phenomena.

We have modified the text to clarify this point to the best of our understanding.

3. n_c seems to be used with two different definitions: on page 5, section E, 2nd paragraph, I think it implies the number of independent configurations (different rN at the same thermodynamic condition), while on page 3, section B, second paragraph it seems to mean the number of replicates in the isoconfigurational ensemble used by the authors (same rN , same thermodynamic conditions, distinct velocity initializations).

Reply: In both cases (pages 3 and 5), we used n_c with the same meaning, *i.e.* the number of replicates in the isoconfigurational ensemble. The number of independent configurations (different rN at the same thermodynamic condition) is referred to as n_g .

While the distinction between both notations is made in the methods section of the article, we understand that the main text was not clear enough to determine which is meant. We modified the main text to clarify this issue.

4. The citations are heavily biased towards self-citations. Things like swap MC have a long history of application, and the isoconfigurational ensemble approach used by the authors was developed by Harrowell and co-workers. The authors should at least credit the pioneers of these methods.

Reply: We thank the Reviewer for raising this issue (see also the answer to Reviewer 2). We have carefully reconsidered all References in the revised manuscript. We have considerably reduced the number of self-citations. We hope that in its revised version, the manuscript credits correctly previous works (keeping in mind that the total number of references allowed by Nature Communications is limited to 50).

REVIEWERS' COMMENTS:

Reviewer #1 (Remarks to the Author):

The authors have completely addressed all of the comments in my first review. I therefore recommend publication of the manuscript.

Reviewer #2 (Remarks to the Author):

The authors have made an effort to explain their results in a more broader scenario and they have addressed my previous concerns as well.

The manuscript is ready for publication.

I would prefer, though, that my name is not associated to this paper, so that I do not give permission to publish my name as a reviewer, although my reviews can be posted online.

Reviewer #3 (Remarks to the Author):

The authors have thoroughly addressed all of my concerns, and I am happy to now recommend publication. Also, a comment on one of the major concerns of reviewer 2 about the novelty of the work presented herein and whether it merits publication in Nature Communications. I'm receptive to the argument that there is some overlap in methods and results in this work, but I think there is tremendous value in having all of the results explained so carefully in a single, high-profile publication. This, plus the additional work mentioned by the authors in their reply, makes me supportive of this article's publication in Nature Communications.

Robert Riggleman

Reply to Reviewer 1 – NCOMMS-19-18724A

[...] The authors have completely addressed all of the comments in my first review. I therefore recommend publication of the manuscript.

We thank Reviewer 1 for the positive appreciation of our work and the constructive comments made in order to improve our manuscript.

Reply to Reviewer 2 – NCOMMS-19-18724A

The authors have made an effort to explain their results in a more broader scenario and they have addressed my previous concerns as well. The manuscript is ready for publication.

I would prefer, though, that my name is not associated to this paper, so that I do not give permission to publish my name as a reviewer, although my reviews can be posted online.

Reply:

We thank the referee for his/her critical reading of our manuscript, and for suggesting ways to improve its accessibility for a broad audience.

Reply to Reviewer 3 – NCOMMS-19-18724A

[...] The authors have thoroughly addressed all of my concerns, and I am happy to now recommend publication. Also, a comment on one of the major concerns of reviewer 2 about the novelty of the work presented herein and whether it merits publication in Nature Communications. I'm receptive to the argument that there is some overlap in methods and results in this work, but I think there is tremendous value in having all of the results explained so carefully in a single, high-profile publication. This, plus the additional work mentioned by the authors in their reply, makes me supportive of this article's publication in Nature Communications.

We thank Reviewer 3 for the positive review and recommendation to publish the manuscript in Nature Communications.